

# Holographic tensor network for double-scaled SYK

**Kazumi Okuyama**

Department of Physics, Shinshu University, 3-1-1 Asahi, Matsumoto 390-8621, Japan

kazumi@azusa.shinshu-u.ac.jp

## Abstract

We construct a holographic tensor network for the double-scaled SYK model (DSSYK). The moment of the transfer matrix of DSSYK can be mapped to the matrix product state (MPS) of a spin chain. By adding the height direction as a holographic direction, we recast the MPS for DSSYK into the holographic tensor network whose building block is a 4-index tensor with the bond dimension three.

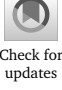
## 1 Introduction

Tensor Network is a useful method to analyze quantum many-body systems (see e.g. [1] for a review). In the context of AdS/CFT correspondence, the holographic tensor network is studied as a toy model for the quantum error correcting property of the holographic duality [2]. However, it is not straightforward to construct a holographic tensor network for a given boundary theory, say the Sachdev-Ye-Kitaev (SYK) model [3–5]. It is desirable to find a concrete example of the holographic tensor network for a known boundary theory.

In this paper, we construct a holographic tensor network for the double-scaled SYK model (DSSYK). As shown in [6], DSSYK is exactly solvable by the technique of the transfer matrix written in terms of the $q$-deformed oscillator. Curiously, it is observed in [7, 8] that the same

transfer matrix also appears in a completely different model, known as the asymmetric simple exclusion process (ASEP) [9] which is a 1d lattice gas model of hopping particles with hard core exclusion. The probability of a configuration of ASEP obeys a Markov process, and its Markov matrix turns out to be a Hamiltonian of the integrable open XXZ spin chain [10] and the stationary state of ASEP is given by a matrix product state (MPS) of the spin chain. Using the correspondence of the transfer matrix of ASEP and DSSYK, we can express the computation of the moment of the transfer matrix of DSSYK as the MPS of a spin chain. Furthermore, when $q = 0$ this MPS turns out to be exactly equal to the ground state of the so-called Fredkin spin chain [11,12]. Recently, the holographic tensor network for the ground state of Fredkin spin chain was constructed in [13]. By generalizing the construction of [13] to the $q \neq 0$ case, we find the holographic tensor network for DSSYK.

This paper is organized as follows. In section 2, we briefly review DSSYK and its exact solution in terms of the $q$-deformed oscillator. In section 3, we explain the relationship between DSSYK, ASEP, and the Fredkin spin chain. In section 4, we construct a holographic tensor network for DSSYK following the approach of [13] with a slight modification. Finally we conclude in section 5 with some discussion of future problems.

## 2 Review of DSSYK

In this section, we will briefly review the definition of double-scaled SYK model (DSSYK) and its solution in terms of the transfer matrix [6]. The Hamiltonian of the SYK model is given by the $p$-body interaction of $N$ Majorana fermions $\psi_i$ $(i = 1, \ldots, N)$

$$H = \mathrm{i}^{p/2} \sum_{1 \leq i_q < \cdots < i_p \leq N} J_{i_1 \cdots i_p} \psi_{i_1} \cdots \psi_{i_p}, \tag{1}$$

where $J_{i_1 \cdots i_p}$ is a Gaussian random coupling with zero mean and the variance

$$\langle J^2_{i_1 \cdots i_p} \rangle = \mathcal{J}^2 \frac{p!(N-p)!}{N!}. \tag{2}$$

For simplicity, we will set $\mathcal{J} = 1$ in what follows. DSSYK is defined by the scaling limit

$$N, p \to \infty, \qquad \text{with} \qquad \lambda = \frac{2p^2}{N}: \quad \text{fixed.} \tag{3}$$

The expectation value of the moment $\langle \mathrm{Tr} H^k \rangle$ averaged over the random coupling $J_{i_1 \cdots i_p}$ can be computed by the Wick contraction since we assumed that the coupling $J_{i_1 \cdots i_p}$ is Gaussian random. For each Wick contraction of $J_{i_1 \cdots i_p}$, we assign a chord called the "$H$-chord". In the scaling limit (3), the remaining trace over the Hilbert space of Majorana fermions boils down to the computation of the intersection number of chords with the weight factor $q = e^{-\lambda}$. This counting problem of intersection numbers of chord diagrams can be solved by introducing the transfer matrix $T$

$$T = a_+ + a_-, \tag{4}$$

where $a_\pm$ is the $q$-deformed oscillator obeying the relation

$$a_- a_+ - q a_+ a_- = 1. \tag{5}$$

Then the moment $\langle \mathrm{Tr} H^k \rangle$ is written as

$$\langle \mathrm{Tr} H^k \rangle = \langle 0 | T^k | 0 \rangle, \tag{6}$$

where $|n\rangle$ $(n = 0, 1, \ldots)$ is the so-called chord number state which represents the state with $n$ chords and $a_\pm$ act as the creation/annihilation operators of chords

$$a_+|n\rangle = \sqrt{\frac{1 - q^{n+1}}{1 - q}}|n + 1\rangle, \qquad a_-|n\rangle = \sqrt{\frac{1 - q^n}{1 - q}}|n - 1\rangle. \tag{7}$$

The 0-chord state is annihilated by $a_-$

$$a_-|0\rangle = 0, \tag{8}$$

and the inner product of chord number states is normalized as $\langle n|m\rangle = \delta_{n,m}$. From (6), the disk partition function $Z(\beta)$ of DSSYK is written as

$$Z(\beta) = \left\langle \mathrm{Tr}\, e^{-\beta H} \right\rangle = \langle 0|e^{-\beta T}|0\rangle. \tag{9}$$

The transfer matrix $T$ is diagonalized in the $|\theta\rangle$-basis

$$T|\theta\rangle = E(\theta)|\theta\rangle, \tag{10}$$

where the eigenvalue $E(\theta)$ is given by

$$E(\theta) = \frac{2\cos\theta}{\sqrt{1 - q}} \qquad (0 \le \theta \le \pi). \tag{11}$$

The overlap of the chord number state $|n\rangle$ and the eigenstate $|\theta\rangle$ of $T$ is given by the $q$-Hermite polynomial $H_n(x|q)$ with degree $n$

$$\langle n|\theta\rangle = \langle \theta|n\rangle = \frac{H_n(\cos\theta|q)}{\sqrt{(q;q)_n}}, \tag{12}$$

which is orthogonal with respect to the measure $\mu(\theta) = (q, e^{\pm 2i\theta}; q)_\infty$

$$\int_0^\pi \frac{d\theta}{2\pi} \mu(\theta) \langle n|\theta\rangle \langle \theta|m\rangle = \langle n|m\rangle = \delta_{n,m}. \tag{13}$$

We can also consider the matter operator $\mathcal{O}_\Delta$ with the dimension $\Delta = s/p$

$$\mathcal{O}_\Delta = \mathrm{i}^{s/2} \sum_{1 \le i_1 < \cdots < i_s \le N} K_{i_1 \cdots i_s} \psi_{i_1} \cdots \psi_{i_s}. \tag{14}$$

Assuming that $K_{i_1 \cdots i_s}$ is another Gaussian random coupling independent of $J_{i_1 \cdots i_p}$, the correlator of $\mathcal{O}_\Delta$'s can be computed by the technique of the chord diagram as well. In this case, there appear two types of chords, the $H$-chord and the matter chord, coming from the Wick contractions of $J$ and $K$, respectively. For instance, the thermal two-point function of the operator $\mathcal{O}_\Delta$ is written as

$$\left\langle \mathrm{Tr}\left[ e^{-\beta_1 H} \mathcal{O}_\Delta e^{-\beta_2 H} \mathcal{O}_\Delta \right] \right\rangle = \langle 0|e^{-\beta_1 T} q^{\Delta \hat{n}} e^{-\beta_2 T}|0\rangle, \tag{15}$$

where $\hat{n}$ is the number operator

$$\hat{n}|n\rangle = n|n\rangle \qquad (n = 0, 1, \ldots). \tag{16}$$

## 3 Spin chain and matrix product state

As mentioned in [7,8], the $q$-oscillator representation of the transfer matrix also appears in a statistical mechanical problem known as the asymmetric simple exclusion process (ASEP) [9]. ASEP is a lattice gas model of particles hopping in a preferred direction with hard core exclusion imposed. The rate for the particle to hop to the left site and the right site is 1 and $q$, respectively.

Let us consider a one-dimensional lattice with $k$ sites where each site can be empty or occupied by a particle. Then, a configuration of the system is specified by a $k$-tuple $\mathcal{C} = (\tau_1, \tau_2, \ldots, \tau_k)$ where $\tau_i = 0$ if the site $i$ is empty and $\tau_i = 1$ if a particle is on site $i$. The probability $P(\mathcal{C})$ for the configuration $\mathcal{C}$ is determined by a Markov process

$$\frac{d}{dt}|P\rangle = M|P\rangle,\tag{17}$$

where $|P\rangle$ is given by

$$|P\rangle = \sum_{\tau_i = 0,1} P(\tau_1, \ldots, \tau_k)|\tau_1, \ldots, \tau_k\rangle.\tag{18}$$

It turns out that the Markov matrix $M$ of ASEP is given by the Hamiltonian of the open XXZ spin chain [10], where we identify $\tau_i = 0, 1$ as the spin up and spin down at site $i$. As shown in [9], the steady state $M|P\rangle = 0$ of ASEP is written as a matrix product state (MPS)

$$|P\rangle = \frac{1}{Z_k} \sum_{\tau_i = 0,1} \langle W| \prod_{i=1}^{k} \big[ \tau_i D + (1 - \tau_i) E \big] |V\rangle |\tau_1, \ldots, \tau_k\rangle,\tag{19}$$

where $D$ and $E$ are written in terms of the $q$-deformed oscillator $a_\pm$ obeying (5)

$$D = \frac{1}{1-q} + \frac{a_-}{\sqrt{1-q}}, \qquad E = \frac{1}{1-q} + \frac{a_+}{\sqrt{1-q}},\tag{20}$$

and $|V\rangle$ and $\langle W|$ are the coherent states of $a_\pm$

$$a_-|V\rangle = v|V\rangle, \qquad \langle W|a_+ = w\langle W|.\tag{21}$$

$Z_k$ in (19) is a normalization factor

$$Z_k = \langle W|(D+E)^k|V\rangle,\tag{22}$$

which ensures the probability interpretation of the coefficient $P(\tau_1, \ldots, \tau_k)$ in (18)

$$\sum_{\tau_i = 0,1} P(\tau_1, \ldots, \tau_k) = 1.\tag{23}$$

If we set $v = w = 0$, $Z_k$ in (22) becomes

$$Z_k = \langle 0|(D+E)^k|0\rangle = \langle 0|\left(\frac{2}{1-q} + \frac{T}{\sqrt{1-q}}\right)^k|0\rangle,\tag{24}$$

where $T$ is the transfer matrix of DSSYK in (4). Thus, the transfer matrix $D + E$ of ASEP is essentially the same as the transfer matrix of DSSYK up to a constant shift and the change of normalization, and $Z_k$ of ASEP is written as a combination of the moments $\langle 0|T^j|0\rangle$ ($j \le k$) of DSSYK. From (11), one can see that this constant shift makes the spectrum of $D + E$ positive definite. For general $v$ and $w$, the boundary states $|V\rangle$ and $\langle W|$ correspond to the end of the world branes in DSSYK [8].

From this relationship between ASEP and DSSYK, it is natural to introduce the (un-normalized) spin chain state for DSSYK

$$|\Psi_k\rangle = \sum_{s_i=\pm} \langle 0|a_{s_1}\cdots a_{s_k}|0\rangle |s_1,\ldots,s_k\rangle, \tag{25}$$

associated with the expansion of the moment $\langle 0|T^k|0\rangle$

$$\langle 0|T^k|0\rangle = \sum_{s_i=\pm} \langle 0|a_{s_1}\cdots a_{s_k}|0\rangle. \tag{26}$$

This moment vanishes for odd $k$, and hence without loss of generality we can assume that $k$ is even

$$k = 2\ell \qquad (\ell \in \mathbb{Z}_{>0}). \tag{27}$$

By expressing $a_\pm$ as 45-degree arrows

$$a_- \sim \; \nearrow, \qquad a_+ \sim \; \searrow, \tag{28}$$

the non-zero coefficients of $|\Psi_k\rangle$ in (25) can be mapped to the so-called Dyck paths of length $k$. For instance, when $k = 4$ there are two types of Dyck path

$$\langle 0|a_-a_-a_+a_+|0\rangle = \quad\diagup\diagup\diagdown\diagdown \quad, $$
$$\langle 0|a_-a_+a_-a_+|0\rangle = \quad\diagup\diagdown\diagup\diagdown \quad. \tag{29}$$

In general, the coefficient $\langle 0|a_{s_1}\cdots a_{s_k}|0\rangle$ in (25) depends on $q$. When $q = 0$, the non-zero coefficients in (25) become 1 for all Dyck paths and the state $|\Psi_k\rangle$ is written as a sum over the Dyck paths with unit coefficient. Interestingly, such a state has been identified as the ground state $|GS\rangle$ of the so-called Fredkin spin chain [11, 12]

$$|GS\rangle = \sum_{(s_1,\ldots,s_k)\in D_k} |s_1,\ldots,s_k\rangle, \tag{30}$$

where $D_k$ denotes the set of Dyck paths of length $k$. As we explained above, the ground state of the Fredkin spin chain in (30) is the $q \to 0$ limit of the MPS for DSSYK in (25)

$$\lim_{q\to 0} |\Psi_k\rangle = |GS\rangle. \tag{31}$$

## 4 Tensor network for DSSYK

In this section, we consider a tensor network representation of the MPS for DSSYK $|\Psi_k\rangle$ in (25). MPS itself is an example of tensor network with a single layer. However, the MPS $|\Psi_k\rangle$ in (25) is not a simple representation of the state since the bond dimension is large. The coefficient $\langle 0|a_{s_1}\cdots a_{s_k}|0\rangle$ of $|\Psi_k\rangle$ in (25) is given by the matrix element of the $q$-deformed oscillator $a_\pm$, which acts on the so-called chord Hilbert space

$$\mathcal{H} = \bigoplus_{n=0}^{\infty} \mathbb{C}|n\rangle. \tag{32}$$

The bond dimension of the MPS $|\Psi_k\rangle$ in (25) is given by $\dim\mathcal{H}$, which is infinite.

As we will see below, we can recast $|\Psi_k\rangle$ in (25) into the network of 4-index tensors with a finite bond dimension by adding an extra direction to make the single layer MPS to a multi-layered tensor network. We interpret the added direction as the holographic direction. We closely follow the construction of the holographic tensor network for the ground state of Fredkin spin chain in [13],[1] but the detail of our construction is slightly different from [13].

To construct the holographic tensor network of $|\Psi_k\rangle$ in (25), we use the bijection between the Dyck paths and the non-crossing pairings (see [16] for a nice review of this bijection). Let us explain this bijection using the example of Dyck paths of length 4 in (29). We draw a vertical line upward from the bottom of the leftmost segment of the Dyck path. When this vertical line hits the path, we draw a horizontal line to the right. When the horizontal line hits the path, we draw a vertical line downward. We repeat this process for the segment of Dyck path which is not hit by the vertical/horizontal lines until all the segments are connected by some vertical/horizontal lines. For the Dyck paths in (29), the resulting vertical/horizontal lines (colored green) become

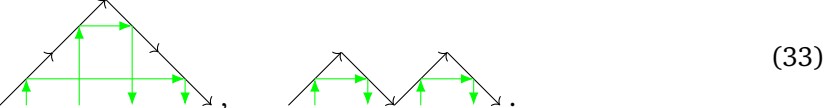

$$(33)$$

They correspond to non-crossing pairings of four sites

$$\overline{1\ 2\ 3\ 4}\ ,\qquad \overline{1\ 2}\ \overline{3\ 4}\ . \tag{34}$$

Note that the crossed pairing $1\ 2\ 3\ 4$ is excluded. The above non-crossing pairings can be thought of as the chord diagram of DSSYK at $q=0$. As we will see below, the $q$-dependence can be incorporated into the value of the tensors.

Next, we introduce the tiling of Dyck path. We draw (a part of) the square lattice in such a way that the end-points of the segment of Dyck path match the vertices of the lattice. For the Dyck paths in (34), the tiling (colored red) is given by

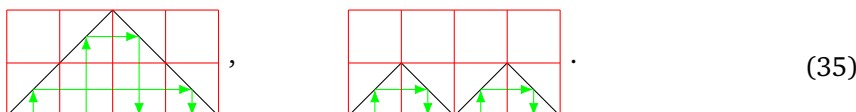

$$(35)$$

We assign the labels $i=\pm,0$ for each edge of the tile: $i=+$ for the head of the green arrow, $i=-$ for the tail of the green arrow, and $i=0$ if the green arrow does not touch the edge. From (35), one can see that there are five types of tiles

$$(36)$$

$$t_1 = \ 0\ \boxed{\phantom{x}}\ +\ ,\qquad t_2 = \ -\ \boxed{\phantom{x}}\ 0\ ,$$

$$t_3 = \ -\ \boxed{\phantom{x}}\ +\ ,\qquad t_4 = \ -\ \boxed{\phantom{x}}\ +\ ,\qquad t_5 = \ 0\ \boxed{\phantom{x}}\ 0\ .$$

---

[1]See also [14,15] for the holographic tensor network of the ground state of (colored) Motzkin spin chain.

The above $t_a$ $(a = 1, \ldots, 5)$ can be thought of as a 4-index tensor

$$i_4 - \boxed{t_a} - i_2 \,, \qquad (37)$$

where the tensor indices $i_1, i_2, i_3, i_4$ take three values $\{\pm, 0\}$. In other words, the bond dimension is three. For instance, the tensor structure of $t_1$ in (36) is written as

$$(t_1)_{i_1 i_2 i_3 i_4} = \delta_{i_1,0} \delta_{i_2,+} \delta_{i_3,-} \delta_{i_4,0} \,. \qquad (38)$$

Other $t_a$'s are defined similarly as the product of four Kronecker $\delta$'s.

Following [13], we introduce the notion of valid tilings: we call a tiling valid if the indices of the edge of the adjacent tiles add up to zero for all overlapping edges. Schematically, this condition is depicted as

$$\boxed{t_a \mid t_b} \quad = \sum_{i,i' \in \{\pm, 0\}} \delta_{i+i',0} \quad \boxed{t_a} \overset{i \; i'}{\vert} \boxed{t_b} \,. \qquad (39)$$

We also impose the same condition for the horizontal edges. One can easily see that the tilings in (35) are valid tilings. We can regard (39) as the multiplication rule of our tensors.

Now we are ready to construct the holographic tensor network for DSSYK. In order to reproduce the MPS in (25), we introduce the tensor $A_h$ which depends on the height $h$ of the tile, where $h \in \mathbb{Z}_{>0}$ is defined by the vertical position of the tile

$$\qquad (40)$$

In order to reproduce the matrix element of $a_\pm$ in (7), we define $A_h$ as the superposition of $t_a$ in (36)

$$A_h = \sqrt{\frac{1-q^h}{1-q}}(t_1 + t_2) + t_3 + t_4 + t_5 \,. \qquad (41)$$

Above, the first two terms of (41) represent the matrix elements

$$\langle h-1|a_-|h\rangle = \langle h|a_+|h-1\rangle = \sqrt{\frac{1-q^h}{1-q}} \,. \qquad (42)$$

Then the MPS $|\Psi_k\rangle$ in (25) for $k = 2\ell$ is written as

$$\qquad (43)$$

Namely, the coefficient $\langle 0|a_{s_1}\cdots a_{s_{2\ell}}|0\rangle$ of MPS is written as a rectangular tensor network with the height $\ell$ and the length $2\ell$, where all the indices of the left, right, and top are set to zero while the bottom indices are given by the configuration of spins $(s_1,\ldots,s_{2\ell})$. Note that we can increase the height of the rectangle in (43) from $\ell$ to $\ell'$ ($\ell' > \ell$) without changing the boundary condition for the tensor indices, since only the empty tile $t_5$ in $A_h$ contributes when $h > \ell$.

The tensor network representation (43) of $\langle 0|a_{s_1}\cdots a_{s_{2\ell}}|0\rangle$ can be easily understood from the examples, such as (35). The green arrows only touch the bottom edge of the rectangle and hence the bottom indices correspond to spins, while the other edges of the rectangle are not crossed by the green arrows and thus assigned the index 0. For instance, the matrix element $\langle 0|a_-^2 a_+^2|0\rangle$ is given by

$$
\begin{aligned}
\langle 0|a_-^2 a_+^2|0\rangle &= \langle 0|a_-|1\rangle\langle 1|a_-|2\rangle\langle 2|a_+|1\rangle\langle 1|a_+|0\rangle \\
&= \sqrt{\frac{1-q}{1-q}}\sqrt{\frac{1-q^2}{1-q}}\sqrt{\frac{1-q^2}{1-q}}\sqrt{\frac{1-q}{1-q}} \\
&= 1+q\,.
\end{aligned}
\tag{44}
$$

We can check that (44) is reproduced from the tensor network (43) with the height $h = 2$ and the spins $(s_1,s_2,s_3,s_4) = (-,-,+,+)$. In this case, the non-zero contribution comes from the first diagram of (35) and the $q$-dependence of (44) is reproduced from $A_1$ and $A_2$ in (41). In the chord diagram computation of the moment $\langle \mathrm{tr}\, H^k\rangle$, the last term $q$ of (44) comes from the intersection of two chords. As we explained below (34), the non-crossing pairings correspond to the chord diagrams at $q = 0$. However, we should stress that when $q \neq 0$ the set of non-crossing pairings in (34) is not equal to the set of chord diagrams, since the crossed pairing is excluded in (34). Nevertheless, the $q$-dependence of $\langle 0|a_{s_1}\cdots a_{s_{2\ell}}|0\rangle$ is correctly reproduced from our tensor network (43) as we have seen for the example $\langle 0|a_-^2 a_+^2|0\rangle$. In other words, the non-crossing pairings do not directly correspond to the chord diagrams when $q \neq 0$, and hence our tensor network does not have a direct interpretation in terms of the chord diagrams. Rather, we constructed our tensor $A_h$ in (41) in such a way that the matrix elements (42) of the $q$-deformed oscillator $a_\pm$ are correctly reproduced.

In the $q \to 0$ limit, $A_h$ in (41) becomes $h$-independent

$$
\lim_{q\to 0} A_h = \sum_{a=1}^{5} t_a\,,
\tag{45}
$$

and our tensor network reduces to the holographic tensor network for the ground state of Fredkin spin chain. Note that our tensor network is slightly different from [13]: the tensor network in [13] is given by the inverted pyramid, while our tensor network becomes a normal pyramid if we remove the empty tiles $t_5$ (see (35) as an example). This difference is not important for the Fredkin spin chain since the building block of the network is height independent, as shown in (45). However, this difference matters when $q \neq 0$ and we believe that our definition of the holographic tensor network is better suited for the $q \neq 0$ case.

We can generalize our construction of holographic tensor network to the computation of the matter two-point function. The two-point function in (15) is expanded as

$$
\begin{aligned}
\langle 0|e^{-\beta_1 T} q^{\Delta\widehat{n}} e^{-\beta_2 T}|0\rangle &= \sum_{n=0}^{\infty} q^{\Delta n}\langle 0|e^{-\beta_1 T}|n\rangle\langle n|e^{-\beta_2 T}|0\rangle \\
&= \sum_{n,k,l=0}^{\infty} q^{\Delta n}\frac{(-\beta_1)^k(-\beta_2)^l}{k!l!}\langle 0|T^k|n\rangle\langle n|T^l|0\rangle\,.
\end{aligned}
\tag{46}
$$

Thus, to study the matter two-point function, it is sufficient to construct the tensor network for the moment $\langle 0|T^k|n\rangle$. Note that $\langle 0|T^k|n\rangle$ is non-zero if $k \geq n$ and $k + n$ is even. This moment $\langle 0|T^k|n\rangle$ is expanded as

$$\langle 0|T^k|n\rangle = \sum_{s_i=\pm} \langle 0|a_{s_1}\cdots a_{s_k}|n\rangle. \tag{47}$$

In a similar manner as above, the matrix element $\langle 0|a_{s_1}\cdots a_{s_k}|n\rangle$ has a tensor network representation

$$
\begin{array}{ccccccc}
& 0 & 0 & \cdots & & 0 & 0 \\
0 & A_h & A_h & \cdots \cdots & & A_h & A_h & 0 \\
\vdots & \vdots & \vdots & & & \vdots & \vdots & 0 \\
0 & A_n & A_n & \cdots \cdots & & A_n & A_n & + \\
\vdots & \vdots & \vdots & & & \vdots & \vdots & \vdots \\
0 & A_1 & A_1 & \cdots \cdots & & A_1 & A_1 & + \\
& s_1 & s_2 & \cdots & & s_{k-1} & s_k
\end{array}
\tag{48}
$$

The boundary condition is changed from (43), where the first $n$ indices on the right side are replaced by $+$ instead of $0$. The height $h$ of the rectangle in (48) should satisfy

$$h \geq \frac{k+n}{2}. \tag{49}$$

## 5 Discussion

In this paper, we have constructed a holographic tensor network for DSSYK. Using the bijection between the Dyck paths and the non-crossing pairings, we can define the tiling of the Dyck path, from which we immediately read off the tensor network with five basic building blocks $\{t_a\}_{a=1,\dots,5}$. By adding the height direction as a holographic direction, we can express the MPS $|\Psi_k\rangle$ for DSSYK in terms of the holographic tensor network made out of the height-dependent tensors $\{A_h\}_{h=1,2,\dots}$, where $A_h$ is a linear combination of $t_a$'s. $A_h$ is a 4-index tensor with bond dimension three, which improves the original MPS expression with the infinite bond dimension.

There are many interesting open questions. Here we list some of them:

- **Entanglement entropy**
  The entanglement entropy of the ground state of Fredkin spin chain was computed in [11]. For instance, if we divide the $k = 2\ell$ spins into two subsystems with $\ell$ spins, the entanglement entropy of $|\text{GS}\rangle$ in (30) scales as

  $$S = \frac{1}{2}\log\ell + \mathcal{O}(1) \qquad (\ell \gg 1). \tag{50}$$

  It would be interesting to study the entanglement entropy of the MPS $|\Psi_k\rangle$ in (25) for $q \neq 0$ and see if the Ryu-Takayanagi formula [17] holds for our holographic tensor network.

- **Asymptotic behavior of Dyck paths**
  We are interested in the large $k$ behavior of the MPS $|\Psi_k\rangle$ in (25). This problem is closely related to the large $k$ asymptotics of the length $k$ Dyck paths or the non-crossing pairings, which have been studied in the literature before [16, 18, 19]. As discussed in [20], the limit shape of the Dyck paths can be thought of as the trajectory of boundary particle in

the bulk spacetime. It would be interesting to compute the limit shape for $q \neq 0$ along the lines of [21, 22].

- **Semi-classical limit of the holographic tensor network**
  It is expected that the $q \to 1$ or the $\lambda \to 0$ limit correspond to the semi-classical limit of the bulk quantum gravity. As discussed in [8, 20, 23, 24], the bulk geodesic length of DSSYK is discretized in units of $\lambda$. In our holographic tensor network, $\lambda$ can be thought of as the lattice spacing in the height direction and the $\lambda \to 0$ limit defines a continuum limit of the tensor network. It would be interesting to take the continuum limit of our holographic tensor network and see if it has some relation to the continuous MERA [25].

- **Holographic error correcting code**
  It is argued in [26] that the AdS/CFT correspondence is an example of the quantum error correcting code, and the toy model of holographic tensor network with this property was proposed in [2]. It would be interesting to see if our holographic tensor network for DSSYK has the property of the quantum error correcting code.

- **Baby universe operator**
  We have seen that the matter operator can be realized in the holographic tensor network by changing the boundary condition for the tensor indices. It would be interesting to consider more exotic operator, like the baby universe operator in DSSYK [27], and see if it has a simple representation in the holographic tensor network.

- **Tensor network for ETH matrix model**
  As discussed in [24], we can construct a matrix model, the so-called ETH matrix model, which reproduces the disk density of states $\mu(\theta)$ of DSSYK in the large $N$ limit. If we ignore the effect of matter operators, the ETH matrix model is given by the $L \times L$ Hermitian one-matrix model

$$\mathcal{Z} = \int_{L \times L} dH e^{-\text{Tr} V(H)}, \tag{51}$$

where $L = 2^{N/2}$ is the dimension of the Hilbert space of $N$ Majorana fermions and the explicit form of the potential $V(H)$ was obtained in [24]. In the ETH matrix model, the average of the moment $\text{Tr} H^k$ is written as

$$\langle \text{Tr} H^k \rangle = \sum_{m=0}^{L-1} \langle m | Q^k | m \rangle, \tag{52}$$

where $Q$ is the so-called Jacobi matrix and the state $|m\rangle$ corresponds to the $m^{\text{th}}$ order orthogonal polynomial with respect to the measure $e^{-V(E)}$ (see [28] for a review). For the even potential $V(-H) = V(H)$, the Jacobi matrix is a tri-diagonal matrix with vanishing diagonal elements, and hence the moment (52) is written as a sum over the Dyck paths. It would be interesting to construct the holographic tensor network for the ETH matrix model at finite $L$, by generalizing our construction at the disk level. At finite $L$, we expect that the holographic tensor network contains the effect of higher genus topologies and it defines the path integral over the fluctuating geometries. It would be interesting to see if the holographic tensor network for the ETH matrix model has some connection to the random tensor network [29].

- **Matter chords and multi-particle chord Hilbert space**
  In this paper, we have primarily focused on the no-particle sector of the chord Hilbert space. However, recent developments have explored the inclusion of matter chords into the chord Hilbert space [30–33]. It would be interesting to construct a tensor network for the multi-particle sector of the chord Hilbert space.

- **Holographic dual of DSSYK**
  It is argued in [34, 35] that DSSYK is holographically dual to the sine dilaton gravity (see also [36, 37]). As we mentioned above, the bulk length of DSSYK is quantized in units of $\lambda$ and hence it is expected that the bulk geometry of DSSYK has some discrete structure. It would be nice if our tensor network can shed light on the holographic dual of DSSYK and the bulk discrete geometry.

Clearly, more work needs to be done to better understand the holographic tensor network for DSSYK. We hope to return to the above questions in the near future.

## Acknowledgments

The author is grateful to Kotaro Tamaoka and Masataka Watanabe for discussions during the "workshop on Low-dimensional Gravity and SYK model" at Shinshu University in March 24–28, 2025.

**Funding information** This work was supported in part by JSPS Grant-in-Aid for Transformative Research Areas (A) "Extreme Universe" 21H05187 and JSPS KAKENHI 22K03594.

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
