# Peer review of "Holographic tensor network for double-scaled SYK"

_SciPost Physics, doi:SciPost Phys. 19, 083 (2025)_

## Round 2 · Referee Report · Anonymous (Referee 1) · 2025-7-30

Strengths

  1. Clear presentation
  2. Intruiging explicit connection between DSSYK and a tensor network

Weaknesses

  1. No holographic interpretation of tensor network is explained
  2. Proofs of key equations are not presented

Report

The explicit realization of matrix elements of the transfer matrix and other operators in DSSYK as a tensor network is very interesting. It opens up the potential to make some of the famous statements about the direct gravitational interpretation of tensor networks (in the context of AdS/CFT) precise (or to explain in which sense they are not precise, we will see). The presentation of the paper is very nice and clear.

I have a few minor remarks 1. Equation (4.11) is one of the main equations, probably the most important one. It's validity is not shown via an explicit calculation. I am not doubting whether the equation is correct, but it may be useful to present a few more details, thus saving the reader some work. The paper is short and pedagogical, so I do not think that a few extra lines would disrupt the work as a whole too much, personally. 2. There is not much "AdS" in this work. It would have been nice to have attempts at relating the tensor network directly gravitational descriptions of DSSYK. Can we literally view the tensor network as building up bulk spacetime? Is the number of tensors as function of h some indication of the size of the bulk spacetime as we go inwards?

These are but very minor remarks, barely complains. I recommend publication.

Requested changes

None.

Recommendation

Publish (easily meets expectations and criteria for this Journal; among top 50%)

  • validity: good
  • significance: top
  • originality: top
  • clarity: top
  • formatting: excellent
  • grammar: acceptable

Author:  Kazumi Okuyama  on 2025-07-31  [id 5695]

(in reply to Report 1 on 2025-07-30)
Category:
remark

  1. Eq.(4.11) is easily understood from the examples, such as eq.(4.4). The green arrows only touch the bottom edge of the rectangle and hence the bottom indices correspond to spins, while the other edges of the rectangle are not crossed by the green arrows and thus assigned the index 0.

  2. AdS interpretation of this tensor network is not clear at the moment. It is a future problem.

---

## Round 2 · Referee Report · Anonymous (Referee 2) · 2025-8-14

Report

See attached file.

Attachment

Recommendation

Publish (meets expectations and criteria for this Journal)

  • validity: -
  • significance: -
  • originality: -
  • clarity: -
  • formatting: -
  • grammar: -

Author:  Kazumi Okuyama  on 2025-08-20  [id 5742]

(in reply to Report 2 on 2025-08-14)
Category:
answer to question

Thank you for carefully reading the manuscript and making valuable comments. Following the suggestion in the referee report, I have made the following changes:

  1. I added eq.(2) for the variance of the random coupling. But we set \mathcal{J}=1 for simplicity.

  2. Z_k is not determined by <P|P>=1, but by the condition eq.(23) coming from the probabilistic interpretation of the Markov process.

  3. I added more explanations on the q-dependence around eq.(44). I stressed that when q is non-zero, the non-crossing pairing is not equal to the chord diagram in general.

  4. I added discussions on the multi-particle chord Hilbert space and the bulk dual of DSSYK at the end of section 5. I cited the references listed in the referee report.

I hope the above changes answer the request in the referee report.

---

## Round 3 · Referee Report · Anonymous (Referee 2) · 2025-9-6

Strengths
1-The presentation is clear and pedagogical.
2-Pointed out a novel correspondence of double-scaled SYK model with a tensor network model, and showed how the partition functions and two-point function can be reproduced from the tensor network side.
2-Pointed out a novel correspondence of double-scaled SYK model with a tensor network model, and showed how the partition functions and two-point function can be reproduced from the tensor network side.
Weaknesses
The extent to which this correspondence holds is unclear. See the report for further details.
Report
Regarding Higher-point correlation functions:
It remains unclear whether higher-point functions, particularly those involving nontrivial topologies or crossing configurations, can be fully reproduced within the tensor network framework. Notably, the appearance of the quantum R-matrix in the crossed four-point function, an essential structure known to encode the chaotic dynamics of DSSYK, has not been addressed in this work.
Semi-classical and Schwarzian limit:
While it is established that DSSYK reduces to JT gravity on the Euclidean disk in the Schwarzian regime, it is natural to ask what the corresponding limit is on the tensor network side. Specifically, how does the tensor network encode the Schwarzian dynamics, and what are the signatures of semi-classical gravitational behavior in its structure? A detailed understanding of this limit would not only shed light on the holographic duality in tensor networks but also offer insights into the emergent geometry and dynamics in the low-energy, semi-classical regime.
It remains unclear whether higher-point functions, particularly those involving nontrivial topologies or crossing configurations, can be fully reproduced within the tensor network framework. Notably, the appearance of the quantum R-matrix in the crossed four-point function, an essential structure known to encode the chaotic dynamics of DSSYK, has not been addressed in this work.
Semi-classical and Schwarzian limit:
While it is established that DSSYK reduces to JT gravity on the Euclidean disk in the Schwarzian regime, it is natural to ask what the corresponding limit is on the tensor network side. Specifically, how does the tensor network encode the Schwarzian dynamics, and what are the signatures of semi-classical gravitational behavior in its structure? A detailed understanding of this limit would not only shed light on the holographic duality in tensor networks but also offer insights into the emergent geometry and dynamics in the low-energy, semi-classical regime.
Recommendation
Publish (easily meets expectations and criteria for this Journal; among top 50%)

Author: Kazumi Okuyama on 2025-09-07 [id 5792]
(in reply to Report 1 on 2025-09-06)The multi-point functions and the JT gravity limit are beyond the scope of this paper. They are future problems.

---

## Editorial Decision

published